# Analyzing the Effect of Baking on the Flavor of Defatted Tiger Nut Flour by E-Tongue, E-Nose and HS-SPME-GC-MS

**DOI:** 10.3390/foods11030446

**Published:** 2022-02-02

**Authors:** Chunbo Guan, Tingting Liu, Quanhong Li, Dawei Wang, Yanrong Zhang

**Affiliations:** 1School of Food Science and Engineering, Jilin Agricultural University, Changchun 130118, China; guanb1112@163.com (C.G.); wangdawei@jlau.edu.cn (D.W.); zhangyanrong0044@jlau.edu.cn (Y.Z.); 2School of Food Science and Nutritional Engineering, China Agricultural University, Beijing 100083, China

**Keywords:** defatted tiger nut flour, flavor, E-tongue, E-nose, HS-SPME-GC-MS, baking

## Abstract

In order to screen for a proper baking condition to improve flavor, in this experiment, we analyzed the effect of baking on the flavor of defatted tiger nut flour by electronic tongue (E-tongue), electronic nose (E-nose) and headspace solid-phase microextraction gas chromatography-mass spectrometry (HS-SPME-GC-MS). According to E-tongue and E-nose radar plots and principal component analysis (PCA), baking can effectively change the taste and odor of defatted tiger nut flour, and the odors of samples with a baking time of >8 min were significantly different from the original odor of unbaked flour. Moreover, bitterness and astringency increased with longer baking times, and sweetness decreased. HS-SPME-GC-MS detected a total of 68 volatile organic compounds (VOCs) in defatted tiger nut flour at different baking levels, and most VOCs were detected at 8 min of baking. Combined with the relative odor activity value (ROAV) and heat map analysis, the types and contents of key flavor compounds were determined to be most abundant at 8 min of baking; 3-methyl butyraldehyde (fruity and sweet), valeraldehyde (almond), hexanal (grassy and fatty), and 1-dodecanol, were the key flavor compounds. 2,5-dimethyl pyrazine, and pyrazine, 2-ethylalkyl-3,5-dimethyl- added nutty aromas, and 1-nonanal, 2-heptanone, octanoic acid, bicyclo [3.1.1]hept-3-en-2-ol,4,6,6-trimethyl-, and 2-pentylfuran added special floral and fruity aromas.

## 1. Introduction

Tiger nut (*Cyperus esculentus* L.), also known as wild chestnut, underground walnut, etc., is a highly efficient and high-quality edible oil crop that is drought-resistant [1]. It originated in North Africa and the Mediterranean, and is widely cultivated in Spain, Italy, South Africa, and other regions [2]. It has high oil content and oil quality, and can be widely planted as a raw material for edible oil [3,4]. As a by-product of tiger nut oil production, defatted tiger nut flour contains many nutrients such as starch, sugar, dietary fiber, protein, vitamins, and minerals [5,6]. It is a raw material for low-fat food with high nutritional value. It can be used to make defatted beverages and pastries that conform to the modern desire for a low-fat and healthy diet. The oil contains much flavor [7], but as most of the oil in the tiger nut is extracted, the fat-soluble flavor compounds are removed in the production of flour, resulting in the reduced flavor of the defatted flour.

Baking is an efficient and rapid means of flavor enhancement. It can make up for the shortcomings of the reduced flavor after defatting, thus meeting consumer preference and improving the breadth of application. During baking, carbonyl compounds and amides trigger the Maillard reaction, Streker degradation, and caramelization reaction, to produce aldehydes, ketones, pyrazines, heterocyclics, etc. [8,9,10]. Too much baking will destroy amino acids and sugars and convert the dominant flavor to a burnt one and cause bitterness [11]. The defatted tiger nut flour contains abundant sugars and amino acids, which can be used as suitable flavor precursors [5,10]. Proper baking condition can retain the flavor of defatted tiger nut flour, increasing the sweet, caramel, and nutty flavors as well as special aromas such as floral and fruity [12]. This makes up for the deficiency of reduced flavor of the flour after defatting, increasing its popularity among consumers and the range of applications. At the same time, baking can sterilize the flour and extend its shelf life, and is thus conducive to its preservation and transportation [13].

E-tongue can digitize taste and effectively distinguish the taste of a sample through the detection of sourness, bitterness, astringency, saltiness, umami, and sweetness [14]. The E-nose utilizes varied “odor fingerprints” for VOCs, and it can precisely identify the differences among various samples [15]. HS-SPME-GC-MS can extract, separate, and detect sample odors and qualitatively and quantitatively assess VOCs [16]. This experiment is the first to analyze the taste and odor of defatted tiger nut flour by E-tongue, E-nose and HS-SPME-GC-MS, and screen for the baking conditions that improve its flavor, thus realizing its high-value utilization as a food raw material. At the same time, with this work, we offer a theoretical basis for the flavor improvement of starchy raw materials such as defatted tiger nut flour, through baking.

## 2. Materials and Methods

### 2.1. Sample Preparation

Tiger nut flour was passed through a 0.42 mm (60-mesh) screen and was added to a tank for supercritical CO_2_ extraction, with the following parameters: extraction pressure 25 MPa, extraction temperature 45 °C, and extraction time 3 h.

The heat of the oven was adjusted to 150 °C and preheated for 10 min. Defatted tiger nut flour was spread 3 mm thick on a baking sheet, placed it the oven, and baked for 0, 4, 8, 12, 16, or 20 min.

### 2.2. HS-SPME

Defatted tiger nut flour (0.8 g) was added to a headspace bottle, which was immediately sealed with a latex cap and equilibrated at 50 °C for 30 min. The SPME fiber (50/30 µm DVB/CAR/PDMS, Supelco, PA, USA) was inserted and pushed out, absorbed for 30 min, slowly retracted, and then immediately inserted in the gas chromatograph inlet, slowly pushed out the fiber head, and desorbed at 250 °C for 5 min. GC-MS analysis was then performed.

### 2.3. GC-MS Analysis

The VOCs were separated in an INNOW-WAX capillary column (30 m × 0.25 mm × 0.25 μm, Agilent J&W, Santa Clara, CA, USA). The temperature conditions were as follows: initial temperature of 40 °C was held for 3 min; raised to 70 °C at a rate of 3 °C/min, held for 2 min; raised to 170 °C at a rate of 5 °C/min, held for 2 min; and raised to 230 °C at a rate of 8 °C/min, then held for 5 min. The carrier gas was helium (99.99%), the flow rate was 1 mL/min, the pressure was 112.0 kPa, the split ratio was 10:1, and the gas chromatograph inlet temperature was 250 °C. The ion source was EI at an energy of 70 eV, the MS transfer line temperature was 230 °C, the ion source temperature was 240 °C, a solvent delay of 3 min was used, and a full scan mode was adopted across the *m*/*z* range of 35 to 500. The VOCs were qualitatively analyzed by NIST mass spectrometry database. Results were accepted when matching and reverse matching values were above 800. Quantitative analysis was based on the area normalization method, removing the erosion of SPME fiber and the capillary column and unidentified peaks. Only the identified peaks were used for normalization.

### 2.4. Relative Odor Activity Value (ROAV)

With reference to Yi C.P. [17], the ROAV method was used to determine the key volatile compounds in defatted tiger nut flour. The compound with the greatest contribution to the sample’s odor was assigned ROAV = 100, and the other compounds’ ROAVs were calculated as follows:$$\mathrm{ROAV}\approx \frac{C_{A}}{C_{stan}}\times \frac{T_{stan}}{T_{A}}\times 100$$
where *C_A_* and *T_A_* are the relative content and threshold of compound A, respectively. *C_stan_, T_stan_* are the relative content and threshold of the compound that contributed the most to the main odor of the sample.

### 2.5. E-Nose Analysis

Defatted tiger nut flour (1 g) was placed into a 20 mL headspace bottle and balanced at (25 ± 1) °C for 0.5 h. The PNE3 electronic nose (PEN3 Airsence, Schwerin, Germany) was used for detection. Table A1 lists the sensor array elements of the E-nose (Table A1, Appendix A). The measurement conditions were as follows: internal flow rate of 400 mL/min, injection flow rate of 100 mL/min, cleaning time of 80 s, sample preparation time of 5 s, and detection time of 60 s.

### 2.6. E-Tongue Analysis

With reference to Cai W.C. [18], analysis was performed on the defatted tiger nut flour using the Taste-Sensing System SA 402B (Intelligent Sensor Technology Co., Ltd., Atsugi, Japan). Table A2 lists the sensor array elements of the E-tongue (Table A2, Appendix B). Defatted tiger nut flour (10 g) was added to 100 mL of 100 °C deionized water and soaked for 30 min. The mixture was centrifuged at 3800× *g* for 20 min at 20 °C, then the supernatant was withdrawn and filtered. The filtrate was used for E-tongue analysis after calibration and diagnosis of the sensor at (25 ± 1) °C. Each sample was tested for six tastes.

### 2.7. Statistical Analysis

E-nose and E-tongue principal component analysis (PCA) and the radar chart were calculated using Origin 2018. Analysis of variance (ANOVA) was used to analyze the differences among samples at a significance level of 0.05 by SPSS 25 (IBM, Armonk, NY, USA). The heat map clustering analysis was performed using the TB tools. All experiments were repeated three times.

## 3. Results

### 3.1. VOCs in Defatted Tiger Nut Flour

Reasonable baking can effectively improve the flavor of defatted tiger nut flour. Some studies have shown that the Maillard reaction temperature is around 100 °C and the caramelization reaction temperature is around 120 °C [19]. If the baking temperature is too low, the material will take a long time to bake, resulting in low baking efficiency and high processing energy consumption. If the baking temperature is too high, it will destroy nutrients such as amino acids and sugars. Furthermore, the reaction rate will be too fast and difficult to control during processing. With overbaking, it is easy for the sample to lose its original odor and develop a burnt odor and bitterness [20]. Therefore, in this experiment, we controlled the baking temperature at 150 °C. It can be seen from Table 1, Figure A1 (Appendix C) and Table A3 (Appendix D), a total of 68 volatile flavor compounds were detected in defatted tiger nut flour with six different baking times. There were 7 aldehydes, 15 alcohols, 6 ketones, 15 esters, 7 acids, 2 olefins, 2 alkanes, 5 pyrazines, and 9 other compounds. The volatile flavor compounds of the baked defatted tiger nut flour increased significantly in types and richness, but varied significantly with different baking times. After baking, the relative content of hexanal and 1-hexanol decreased, indicating that grassy and fruity aromas decreased [21]. Furthermore, the relative content of ketones and pyrazines increased, indicating that flowery, sweet, and nutty aromas increased [22].

Aldehydes are mainly produced by lipid oxidation, decomposition, and Strecker degradation of amino acid, and their odor threshold is low [23,24]. They account for a large proportion of the VOCs of defatted tiger nut flour and are the main compounds that affect its odor. Among these, 3-methylbutyraldehyde, valeraldehyde, and hexanal play an essential role in the odor of defatted tiger nut flour. As the baking time increased, 3-methylbutanal was produced by the degradation of leucine, which indicates that cocoa, sweet and baked aromas increased [25]. The reduction in hexanal indicates that the aromas of soybean, malt and grass were reduced; these are oxidation products of unsaturated fatty acids such as linoleic acid and oleic acid [26].

Alcohols, mainly derived from the thermal oxidation of lipids and degradation of carbohydrates [27], with vegetal and aromatic odors, had a higher odor threshold and less impact on the aroma of the defatted tiger nut flour than aldehydes and ketones [28]. Baking increased the types of alcohols; among them, hexanol had a grassy aroma and disappeared after 4 min of baking. Defatted tiger nut flour baked for 8 min produced 1-dodecanol and bicyclo [3.1.1]hept-3-en-2-ol,4,6,6-trimethyl-, which increased violet and verbena aromas. These added new VOCs to the defatted tiger nut flour and improved the richness of the aroma compounds.

Ketones, mainly from the thermal degradation of amino acids or thermal oxidation of polyunsaturated fatty acids [29], have a high odor threshold and impacted odor only slightly [30]. 2-Nonanone and 2-tridecone were formed after baking defatted tiger nut flour, adding fruity and coconut aromas. Upon baking, the relative content of 2-heptanone and 2-decanone increased, indicating that cheese and fruit aromas increased; 2-heptanone is produced by amino acid decomposition [26].

Esters are typically generated by the dehydration of hydroxy fatty acids [31], with a low odor threshold. They have creamy and fruity aromas and are essential components of an odor [32]. Baking defatted tiger nut flour generated γ-undecanolactone, and its relative content gradually increased, enhancing peanut and nutty aromas. γ-Undecanolide (peach aldehyde) was the aroma of the raw material itself, and its relative content rose after baking for 8 min, with coconut and peach-like aromas [33]. γ-Butyrolactone is formed after roasting and has a nutty aroma, resulting from the esterification of hydroxy fatty acids or the oxidation of unsaturated aldehydes.

As one of the critical products of the Maillard reaction, pyrazines are formed by Strecker degradation of leucine, isoleucine, and glycine [34], and have strong sensory characteristics. They typically have baked aromas such as baked hazelnuts, baked barley, and baked corn [35]. Defatted tiger nut flour generated 2-methylpyrazine, 2,5-dimethyl pyrazine, pyrazine,2-methyl-3-(2-methylpropyl)-2,3,5-Trimethylpyrazine, and pyrazine,2-ethyl-3,5- dimethyl- after baking, with typical baking aromas such as fried peanuts, nut, roasted potato, and coffee [36,37]. Furans are oxygenated heterocyclic compounds generated by the Maillard reaction and caramelization, mainly from the cyclization and dehydration of Amadori compounds [25,38], which mainly have caramel and nutty aromas. After baking, the relative content of 2-pentylfuran increased significantly, and buttery, mung bean, floral, and fruity aromas gradually became stronger in the defatted tiger nut flour [39].

Acids are produced by the further oxidation of aldehydes; they have a high odor threshold and have little effect on the odor [40]. They have cheesy, fruity, and sour aromas [41]. When defatted tiger nut flour was baked for 8 min, the relative content of hexanoic acid, octanoic acid, and nonanoic acid increased, indicating increased coconut and cheese aromas. Alkanes have a higher odor threshold and weaker effect on the odor. Olefins generally have a lower odor threshold and present floral and fruity aromas [42]. The higher relative content of (-)-limonene in defatted tiger nut flour at 8 min of baking increased the lemon aroma [43], resulting in richer VOCs.

In summary, reasonable baking conditions can improve the richness of VOCs of defatted tiger nut flour and achieve the purpose of improving aroma. When baked for 8 min, the relative content of valeraldehyde, hexanal, and 1-hexanol in defatted tiger nut flour decreased, and thus, grassy odors decreased. Meanwhile, 3-methylbutyraldehyde, 2-heptanone, 2-nonanone, γ-undecanolide, (-)-limonene and other fruit aroma compounds as well as baking aroma compounds, such as pyrazine, increased. However, excessive baking will destroy the prominent odor and produce a burnt odor and harmful substances. When the baking time exceeded 12 min, pyrazine significantly increased, and the samples had caramel and burnt aromas and bitterness.

As shown in Figure 1, the defatted tiger nut flour primarily contained more aldehydes, followed by esters, alcohols and ketones. Baking for 8 min produced the most abundant VOCs, with 39 types in total, 11 more than the VOCs in unbaked defatted tiger nut flour. There were 15 aldehydes and 13 esters, which were the main VOCs. The baked defatted tiger nut flour produced pyrazines, adding caramel and baked nut aromas. Upon baking for 20 min, the relative content and types of pyrazine were higher, and the content of aldehydes was lower, indicating that the odor had changed too much and that the original and baking aromas had declined while the burnt odor was higher.

### 3.2. Heat Map Analysis of Volatile Flavor Compounds in Defatted Tiger Nut Flour

In the heat map, the difference between the content of various VOCs and the average content is indicated by different shades of color, providing a more visual indication of the differences between the samples [44]. The relative content of each volatile flavor compound is marked with a different color in the heat map. The darker the red, the greater the relative content, and the darker the blue, the less the relative content [45]. As can be seen from Figure 2, when the defatted tiger nut flour was baked for 8 min, the relative contents of 2-pentylfuran (buttery and floral), caprylic acid (cheese), nonanoic acid (fatty and coconut), γ-undecanolactone (peach, coconut, and milk), and bicyclo[3.1.1]hept-3-en-2-ol,4,6,6-trimethyl-(verbena) were significantly higher. When baked for 8 min, the defatted tiger nut flour had more positive VOCs and the relative content was higher, indicating that the richness of its aroma was greater. When baked for 16 and 20 min, the VOCs of defatted tiger nut flour were mainly composed of pyrazines, with coffee and burnt aromas. At this point, the richness of the aromas decreased significantly and their sensory quality was low.

### 3.3. Relative Odor Activity Value (ROAV)

The sample’s odor was determined by both the relative content of each VOC and its threshold value; when the content was certain and the threshold value was smaller, the greater the contribution to odor. The ROAV method was used to determine the contribution of each volatile compound to the main odor. The greater the ROAV value, the greater the contribution to the main odor, and the compounds with ROAV > 1 were key volatile compounds.

It can be seen from Table 2 and Figure 3. Without baking, defatted tiger nut flour’s odor was mainly attributable to 3-methylbutanal, hexanal and 2-heptanone with light fruit, malt, almond, grassy, and other original aromas. When baked for 8 min, its key volatile compound types were 3-methylbutanal, 1-hexanal, octanal, 1-nonanal, 2,5 dimethyl pyrazine, and pyrazine,2-ethyl-3,5-dimethyl-, significantly contributing to the odor. Bicyclo[3.1.1]hept-3-en-2-ol,4,6,6-trimethyl-, 2-heptanone, octanoic acid, (-)-limonene, and other compounds contributed less, indicating that baking for 8 min both retained the original aroma of the defatted tiger nut flour and added fruity, sweet, fried peanut and baking aromas. When baked for 16 and 20 min, the ROAV of 2-ethyl-3,5-dimethyl pyrazine reached 43.18 and 41.62, respectively; thus, the fried peanut and coffee aromas were too heavy, and the samples had a strong burnt odor, which greatly reduced the quality.

### 3.4. E-Nose Analysis of Defatted Tiger Nut Flour

It can be seen from Figure 4A that the profile and area of the radar plot had some variation. The response intensity of the sensors W1C, W5C, W3S, W2W, and W3C was higher with slight variation, indicating that the contents of the aromatic compounds, benzene, ammonia, olefin, short-chain aromatic compounds, organic sulfides, and long-chain alkanes in the samples were high, but the changes were not significant. The sensors W2S, W1W, W6S, W1S and W5S changed significantly, indicating that baking had a significant effect on alcohols, aldehydes, ketones, sulfur compounds, hydrogen, nitrogen oxides and methyl components. This was related to the results of the above VOCs—the relative contents of alcohols, aldehydes, ketones, esters, and pyrazines were greater, and the changes were more significant.

The PCA was performed to further analyze the odor differences of defatted tiger nut flour with different baking times. It can be seen from Figure 4B that the contributions of PC1 and PC2 at different baking times were 77.7% and 14.9%, respectively, with a cumulative contribution of 92.6%, indicating that these two principal components could reflect most of the information of the samples. The defatted tiger nut flour without baking and with baking for 4 and 8 min were relatively close. For the sample baked for more than 8 min, the radar plot distances were further from those of the unbaked sample, indicating that baking drastically changed the original odor of the defatted tiger nut flour. The samples baked for 16 and 20 min were relatively close, indicating that their odors were similar, and their sensory odors were mainly burnt and coffee aromas.

### 3.5. E-Tongue Analysis of Defatted Tiger Nut Flour

It can be seen from Figure 5 that the contributions of PC1 and PC2 were 44.2% and 28.6%, respectively, with a cumulative contribution of 72.8%, indicating that these two principal components could reflect most of the information of the samples. The plotted values for the unbaked and baked defatted tiger nut flour were far apart in the radar plots, indicating that baking significantly affected its taste. As the level of baking increased, the sensors AE1, C00 and GL1 changed significantly. This is because the sugar and amino acids in defatted tiger nut flour, as the flavor precursors of the Maillard reaction and caramelization, produced a caramel taste or bitterness and VOCs such as alcohols, aldehydes and pyrazines [42]. AAE, CT0 and CA0 showed umami, saltness and sourness, respectively, with slight changes.

The radar plots of defatted tiger nut flour baked for 4 and 8 min were closer, indicating similar tastes. Their plots were far away from that for the unbaked defatted tiger nut flour, indicating that the taste was significantly different after baking. However, overbaking will cause the defatted tiger nut flour to produce unfavorable tastes such as astringency and bitterness, and produce coffee, burnt and other odors. When samples were baked for 12, 16, and 20 min, the respective defatted tiger nut flour radar plots were far away from that for the unbaked defatted tiger nut flour, and the samples began to develop bitterness; the sweetness decreased, and the overall taste worsened. According to the above, the VOCs, E-nose and E-tongue changed, the taste and odor of the defatted tiger nut flour were greatly changed at these baking times—its original flavor was lost, changing into a caramel and burnt flavor, indicating that its flavor decreased.

## 4. Conclusions

Tiger nut, as a high-quality oil-making raw material, will produce a large amount of defatted tiger nut flour after oil extraction [46]. Due to its reduced flavor, it is usually used as fertilizer and feed after oil extraction [47]. However, reasonable baking can effectively improve its flavor, and it can be used as a high-quality food raw material to realize its high-value utilization.

With the increase in baking level, the Maillard reaction, Strecker degradation, and caramelization consumed the sugar and amino acids in defatted tiger nut flour, resulting in a caramel taste or bitterness, while at the same time producing VOCs such as alcohols, aldehydes, pyrazines, etc. When the baking time exceeded 12 min, the main odor changed to coffee and burnt odor, the color of the flour was too dark, and the quality was obviously reduced. When the flour was baked for 8 min, the VOCs and key flavor compounds were the richest. The grass-like VOCs such as hexanal and 1-hexanol were reduced, and the VOCs of fruit, roasted peanuts, nuts, chocolate, milk, and sweet aromas such as γ-undecanolide, 1-dodecanol, (-)-limonene, 2-nonane, 2-tridecone, hexanoic acid, and pyrazines were increased. This indicates that a reasonable baking condition can improve the taste, aroma and flavor richness of defatted tiger nut flour.

This experiment showed that 8 min of baking can enhance the flavor of defatted tiger nut flour to compensate for the loss of flavor after oil extraction. Properly baked defatted tiger nut flour becomes a higher-quality food raw material. As a product that meets the needs of today’s market, defatted tiger nut flour can be used to produce a wide variety of low-fat foods and help achieve the complete and high-value utilization of tiger nut resources.

## Figures and Tables

**Figure 1 foods-11-00446-f001:**
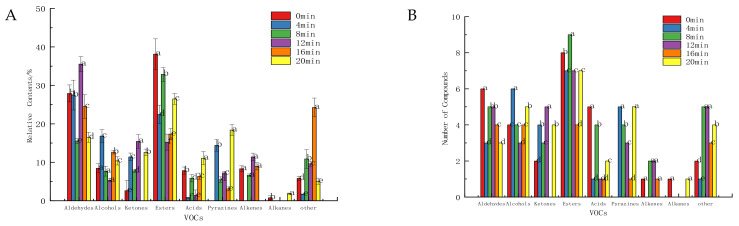
Relative content (**A**) and number (**B**) of VOCs in defatted tiger nut flour. Different lowercase letters in the same classification indicate that there was a significant difference (*p* < 0.05).

**Figure 2 foods-11-00446-f002:**
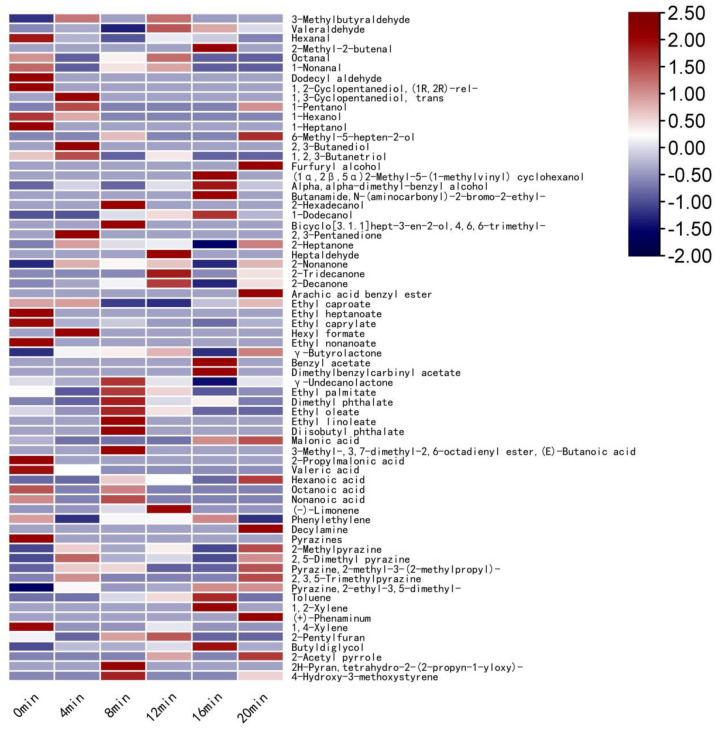
Heat map of volatile compounds in defatted tiger nut flour.

**Figure 3 foods-11-00446-f003:**
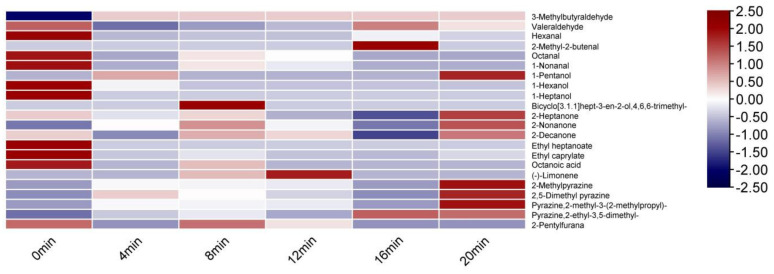
Heat map of ROAV of defatted tiger nut flour.

**Figure 4 foods-11-00446-f004:**
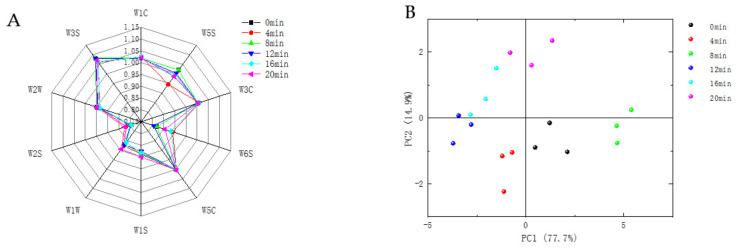
Radar diagram (**A**) and PCA (**B**) of defatted tiger nut flour for E-nose.

**Figure 5 foods-11-00446-f005:**
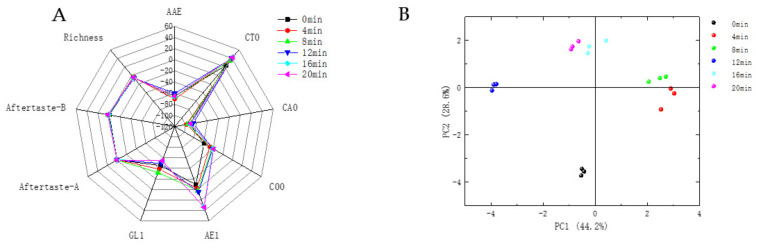
Radar diagram (**A**) and PCA (**B**) of defatted tiger nut flour for E-tongue.

**Table 1 foods-11-00446-t001:** VOCs in defatted tiger nut flour.

Count	RT	Compounds	RI	Formula	Molecular Ion Mass (M+)	CAS#	Relative Content/(%)						Qualitative Method
							0 min	4 min	8 min	12 min	16 min	20 min	
		Aldehydes											
1	3.14	3-Methylbutyraldehyde	652	C_5_H_10_O	86.07	590-86-3	2.65 ± 0.80 ^c^	17.15 ± 1.73 ^a^	7.30 ± 0.25 ^b^	17.32 ± 0.25 ^a^	7.18 ± 0.77 ^b^	7.40 ± 0.77 ^b^	MS, RI
2	4.21	Valeraldehyde	699	C_5_H_10_O	86.07	110-62-3	2.76 ± 0.40 ^cd^	3.07 ± 0.28 ^c^	1.86 ± 0.17 ^d^	5.28 ± 0.65 ^a^	4.55 ± 1.31 ^ab^	3.48 ± 0.29 ^bc^	MS, RI
3	6.88	Hexanal	800	C6H_12_O	100.09	66-25-1	18.03 ± 0.79 ^a^	7.26 ± 1.87 ^cd^	3.99 ± 0.03 ^e^	9.17 ± 0.74 ^b^	8.14 ± 0.83 ^bc^	5.60 ± 0.24 ^de^	MS, RI
4	7.23	2-Methyl-2-butenal	745	C_5_H_8_O	84.06	497-03-0	ND	ND	ND	ND	4.64 ± 0.18 ^a^	ND	MS, RI
5	14.91	Octanal	1003	C_8_H_16_O	128.12	124-13-0	2.09 ± 0.16 ^a^	ND	1.34 ± 0.09 ^b^	2.38 ± 0.28 ^a^	ND	ND	MS, RI
6	19.37	1--Nonanal	1104	C_9_H_18_O	142.13	124-19-6	1.71 ± 0.12 ^a^	ND	1.03 ± 0.07 ^c^	1.38 ± 0.03 ^b^	ND	ND	MS, RI
7	21.60	Dodecyl aldehyde	1409	C_12_H_24_O	184.18	112-54-9	0.66 ± 0.09 ^a^	ND	ND	ND	ND	ND	RI
		Alcohols											
8	7.21	1,2-Cyclopentanediol,(1R,2R)-rel-	-	C_5_H_10_O_2_	102.07	5057-99-8	2.29 ± 0.43 ^a^	ND	ND	ND	ND	ND	MS, RI
9	7.25	1,3-Cyclopentanediol, trans	-	C_5_H_10_O_2_	102.07	16326-98-0	ND	2.14 ± 0.40 ^a^	ND	ND	ND	ND	MS, RI
10	13.41	1-Pentanol	822	C_5_H_12_O	104.07	110-66-7	ND	7.32 ± 0.58 ^a^	ND	ND	ND	5.63 ± 0.30 ^b^	MS, RI
11	18.08	1-Hexanol	868	C_6_H_14_O	102.10	111-27-3	3.46 ± 0.46 ^a^	2.23 ± 0.33 ^b^	ND	ND	ND	ND	MS, RI
12	21.71	1-Heptanol	970	C_7_H_16_O	116.12	111-70-6	1.03 ± 0.10 ^a^	ND	ND	ND	ND	ND	MS, RI
13	23.21	6-Methyl-5-hepten-2-ol	994	C_8_H_16_O	128.12	1569-60-4	ND	ND	0.87 ± 0.15 ^b^	ND	ND	1.54 ± 0.18 ^a^	MS, RI
14	23.99	2,3-Butanediol	-	C_4_H_10_O_2_	90.07	513-85-9	ND	1.85 ± 0.09 ^a^	ND	ND	ND	ND	RI
15	25.43	1,2,3-Butanetriol	-	C_4_H_10_O_3_	106.06	4435-50-1	1.62 ± 0.21 ^b^	2.55 ± 0.25 ^a^	ND	1.37 ± 0.18 ^b^	ND	ND	MS
16	27.60	Furfuryl alcohol	859	C_5_H_6_O_2_	98.04	98-00-0	ND	ND	ND	ND	ND	1.04 ± 0.40 ^a^	MS, RI
17	28.10	(1α,2β,5α)2-Methyl-5-(1-methylvinyl) cyclohexanol	1192	C_10_H_18_O	154.14	38049-26-2	ND	ND	ND	ND	2.92 ± 0.10 ^a^	ND	MS, RI
18	29.84	Alpha,alpha-dimethyl-benzyl alcohol	1090	C_9_H_12_O	136.09	617-94-7	ND	0.74 ± 0.03 ^b^	ND	0.82 ± 0.03 ^b^	2.64 ± 0.18 ^a^	0.65 ± 0.04 ^b^	MS, RI
19	30.11	Butanamide,N-(aminocarbonyl)-2-bromo-2-ethyl-	1521	C_7_H_13_BrN_2_O_2_	236.02	77-65-6	ND	ND	ND	ND	1.33 ± 0.08 ^a^	ND	RI
20	32.19	2-Hexadecanol	1702	C_16_H_34_O	242.26	14852-31-4	ND	ND	3.93 ± 0.50 ^a^	ND	ND	ND	RI
21	34.19	1-Dodecanol	1473	C_12_H_26_O	186.20	112-53-8	ND	ND	2.10 ± 0.41 ^c^	3.17 ± 0.21 ^b^	5.74 ± 0.12 ^a^	1.50 ± 0.14 ^d^	
22	34.47	Bicyclo[3.1.1]hept-3-en-2-ol,4,6,6-trimethyl-	1140	C_10_H_16_O	152.12	473-67-6	ND	ND	0.78 ± 0.21 ^a^	ND	ND	ND	RI
		Ketones											
23	6.38	2,3-Pentanedione	698	C_5_H_8_O_2_	100.05	600-14-6	ND	1.95 ± 0.25 ^a^	ND	ND	ND	ND	MS, RI
24	10.40	2-Heptanone	891	C_7_H_14_O	114.10	110-43-0	1.58 ± 0.62 ^a^	4.35 ± 0.52 ^a^	2.75 ± 0.14 ^b^	2.92 ± 0.41 ^b^	ND	4.73 ± 0.33 ^a^	MS, RI
25	10.54	Heptaldehyde	901	C_7_H_14_O	114.10	111-71-7	ND	ND	ND	1.01 ± 0.14 ^a^	ND	ND	MS, RI
26	19.19	2-Nonanone	1092	C_9_H_18_O	142.14	821-55-6	ND	3.74 ± 0.22 ^a^	2.82 ± 0.14 ^b^	3.45 ± 0.78 ^ab^	ND	3.63 ± 0.26 ^ab^	MS, RI
27	22.70	2-Tridecanone	1497	C_13_H_26_O	198.20	593-08-8	ND	ND	ND	3.70 ± 0.11 ^a^	ND	1.59 ± 0.14 ^b^	MS, RI
28	22.82	2-Decanone	1193	C_10_H_20_O	156.15	693-54-9	1.05 ± 0.21 ^d^	1.36 ± 0.01 ^d^	2.18 ± 0.12 ^c^	4.37 ± 0.32 ^a^	ND	2.64 ± 0.12 ^b^	MS, RI
		Esters											
29	5.67	Arachic acid benzyl ester	3003	C_27_H_46_O_2_	402.35	77509-04-7	ND	ND	ND	ND	ND	4.23 ± 0.16 ^a^	MS
30	12.50	Ethyl caproate	1000	C_8_H_16_O_2_	144.12	123-66-0	15.31 ± 0.28 ^a^	15.53 ± 1.77 ^a^	2.73 ± 0.09 ^c^	2.08 ± 0.18 ^c^	8.56 ± 0.96 ^b^	14.38 ± 0.31 ^a^	MS, RI
31	17.06	Ethyl heptanoate	1097	C_9_H_18_O_2_	158.14	106-30-9	2.12 ± 0.11 ^a^	ND	ND	ND	ND	ND	MS, RI
32	20.87	Ethyl caprylate	1196	C_10_H_20_O_2_	172.14	106-32-1	8.23 ± 1.59 ^a^	1.41 ± 0.07 ^b^	1.94 ± 0.19 ^b^	0.97 ± 0.04 ^b^	ND	1.43 ± 0.21 ^b^	MS, RI
33	21.70	Hexyl formate	1039	C_8_H_16_O_2_	144.12	112-23-2	ND	1.30 ± 0.04 ^a^	ND	ND	ND	ND	RI
34	23.99	Ethyl nonanoate	1296	C_11_H_22_O_2_	186.16	123-29-5	6.35 ± 0.25 ^a^	ND	ND	ND	ND	ND	MS, RI
35	26.56	γ-Butyrolactone	915	C_4_H_6_O_2_	86.04	96-48-0	ND	1.94 ± 0.11 ^c^	2.23 ± 0.43 ^bc^	2.79 ± 0.25 ^ab^	ND	3.33 ± 0.34 ^a^	MS, RI
36	29.04	Benzyl acetate	1164	C_9_H_10_O_2_	150.07	140-11-4	ND	ND	ND	ND	2.71 ± 0.10 ^a^	ND	RI
37	29.95	Arachic acid benzyl ester	1298	C_12_H_16_O_2_	192.12	151-05-3	ND	ND	ND	ND	1.45 ± 0.24 ^a^	ND	MS, RI
38	35.45	γ-Undecanolactone	1576	C_11_H_20_O_2_	184.15	104-67-6	0.68 ± 0.05 ^b^	0.52 ± 0.09 ^b^	1.47 ± 0.09 ^a^	0.71 ± 0.15 ^b^	ND	0.71 ± 0.17 ^b^	MS, RI
39	40.06	Ethyl palmitate	1993	C_18_H_36_O_2_	284.27	628-97-7	2.21 ± 0.11 ^c^	ND	4.75 ± 0.18 ^a^	2.81 ± 0.13 ^b^	ND	0.73 ± 0.06 ^d^	MS, RI
40	40.96	Dimethyl phthalate	1454	C_10_H_10_O_4_	194.06	131-11-3	1.78 ± 1.56 ^d^	1.16 ± 0.14 ^f^	8.83 ± 0.24 ^a^	3.54 ± 0.62 ^c^	4.61 ± 0.13 ^b^	1.68 ± 0.15 ^df^	MS, RI
41	43.36	Ethyl oleate	2173	C_20_H_38_O_2_	310.29	111-62-6	1.45 ± 0.0 ^c^	0.60 ± 0.19 ^d^	4.80 ± 0.10 ^a^	2.35 ± 0.62 ^b^	ND	ND	MS, RI
42	43.96	Ethyl linoleate	2162	C_20_H_36_O_2_	308.27	544-35-4	ND	ND	1.53 ± 0.19 ^a^	ND	ND	ND	MS
43	44.22	Diisobutyl phthalate	2317	C_16_H_22_O_4_	334.21	84-69-5	ND	ND	4.57 ± 0.28 ^a^	ND	ND	ND	MS, RI
		Acids											
44	22.09	Malonic acid	-	C_3_H_4_O_4_	104.01	141-82-2	1.61 ± 0.12 ^b^	ND	ND	ND	6.35 ± 0.83 ^a^	ND	RI
45	25.65	3-Methyl-,3,7-dimethyl-2,6-octadienyl ester,(E)-Butanoic acid	1606	C_15_H_26_O_2_	238.19	109-20-6	ND	ND	1.63 ± 0.22 ^a^		ND	ND	RI
46	27.94	2-Propylmalonic acid	-	C_6_H_10_O_4_	146.06	616-62-6	1.48 ± 0.28 ^a^	ND	ND	ND	ND	ND	MS
47	29.54	Valeric acid	904	C_5_H_10_O_2_	102.07	109-52-4	2.37 ± 0.25 ^a^	0.75 ± 0.13 ^b^	ND	ND	ND	ND	MS, RI
48	31.92	Hexanoic acid	990	C_6_H_12_O_2_	116.08	142-62-1	ND	ND	1.89 ± 0.65 ^b^	1.40 ± 0.18 ^b^	ND	3.23 ± 0.27 ^a^	MS, RI
49	36.37	Octanoic acid	1180	C_8_H_16_O_2_	144.12	124-07-2	1.58 ± 0.05 ^a^	ND	1.33 ± 0.03 ^b^	ND	ND	ND	MS, RI
50	38.75	Nonanoic acid	1273	C_9_H_18_O_2_	158.13	112-05-0	0.79 ± 0.23 ^a^	ND	0.99 ± 0.05 ^a^	ND	ND	ND	MS, RI
		Olefins											
51	10.77	(-)-Limonene	1031	C_10_H_16_	136.13	5989-54-8	ND	ND	1.17 ± 0.24 ^b^	5.93 ± 0.74 ^a^	ND	ND	MS, RI
52	13.38	Phenylethylene	893	C_8_H_8_	104.06	100-42-5	8.31 ± 0.77 ^a^	ND	5.46 ± 0.10 ^b^	5.46 ± 0.25 ^b^	8.95 ± 0.93 ^a^	ND	MS, RI
		Alkanes											
53	18.10	Decylamine	1255	C_10_H_23_N	157.18	2016-57-1	ND	ND	ND	ND	ND	1.81 ± 0.18 ^a^	RI
54	20.68	1,1-Diethoxy-octane	1270	C_12_H_26_O_2_	202.19	54889-48-4	0.78 ± 0.51 ^a^	ND	ND	ND	ND	ND	MS, RI
		Pyrazines											
55	13.73	2-Methylpyrazine	831	C_5_H_6_N_2_	94.05	109-08-0	ND	2.85 ± 0.33 ^b^	1.18 ± 0.05 ^c^	2.48 ± 0.12 ^b^	ND	4.49 ± 0.39 ^a^	MS, RI
56	16.29	2,5-Dimethyl pyrazine	917	C_6_H_8_N_2_	108.07	123-32-0	ND	7.93 ± 0.90 ^a^	2.39 ± 0.54 ^b^	3.36 ± 0.09 ^b^	ND	6.97 ± 0.54 ^a^	MS, RI
57	16.92	Pyrazine,2-methyl-3-(2-methylpropyl)-	1134	C_9_H_14_N_2_	150.12	13925-06-9	ND	0.69 ± 0.09 ^b^	0.65 ± 0.04 ^b^	ND	ND	1.08 ± 0.21 ^a^	MS, RI
58	19.56	2,3,5-Trimethylpyrazine	1004	C_7_H_10_N_2_	122.08	14667-55-1	ND	0.74 ± 0.07 ^b^	ND	ND	ND	0.95 ± 0.07 ^a^	MS, RI
59	21.02	Pyrazine,2-ethyl-3,5-dimethyl	1084	C_8_H_12_N_2_	136.10	13925-07-0	ND	2.20 ± 0.14 ^b^	1.27 ± 0.12 ^c^	1.41 ± 0.07 ^c^	3.10 ± 0.52 ^a^	3.08 ± 0.20 ^a^	MS, RI
		Other											
60	5.63	Toluene	763	C_7_H_8_	92.06	108-88-3	ND	ND	2.44 ± 1.22 ^b^	4.29 ± 0.36 ^a^	8.58 ± 1.58 ^b^	ND	RI
61	8.28	1,2-Xylene	887	C_8_H_10_	106.08	95-47-6	ND	ND	ND	ND	9.21 ± 0.55 ^a^	ND	MS
62	8.34	( + )-Phenaminum	1141	C_9_H_13_N	135.10	51-64-9	ND	ND	ND	ND	ND	1.30 ± 0.16 ^a^	MS
63	8.74	1,4-Xylene	865	C_8_H_10_	106.08	106-42-3	5.21 ± 0.49 ^b^	ND	ND	1.22 ± 0.08 ^c^	ND	ND	MS, RI
64	12.40	2-Pentylfuran	993	C_9_H_14_O	138.10	3777-69-3	0.61 ± 0.08 ^c^	ND	1.08 ± 0.04 ^b^	1.39 ± 0.06 ^a^	ND	ND	MS, RI
65	30.52	Butyldiglycol	1192	C_8_H_18_O_3_	162.13	112-34-5	ND	1.71 ± 0.04 ^b^	1.48 ± 0.81 ^b^	2.21 ± 0.16 ^b^	6.46 ± 0.32 ^a^	1.44 ± 0.14 ^b^	MS, RI
66	35.27	2-Acetyl pyrrole	1064	C_6_H_7_NO	109.05	1072-83-9	ND	ND	ND	0.50 ± 0.02 ^a^	ND	0.75 ± 0.23 ^a^	RI
67	35.73	2H-Pyran,tetrahydro -2-(2-propyn-1-yloxy)-	976	C_8_H_12_O_2_	140.08	6089-04-9	ND	ND	2.66 ± 0.15 ^a^	ND	ND	ND	MS, RI
68	39.27	4-Hydroxy-3-methoxystyrene	1317	C_9_H_10_O_2_	150.07	7786-61-0	ND	ND	3.21 ± 0.22 ^a^	ND	ND	1.53 ± 0.22 ^b^	MS, RI

”ND”: volatile flavor compounds not detected. Different lowercase letters in the same row indicate that there was a significant difference (*p* < 0.05). MS: Identification based on the NIST mass spectrometry database.

**Table 2 foods-11-00446-t002:** ROAV of defatted tiger nut flour.

Count	Compounds	Aroma Characteristics	Threshold μg/kg	ROAV					
				0 min	4 min	8 min	12 min	16 min	20 min
1	3-Methylbutyraldehyde	Light Fruit, Sweet, Malt	1.00	66.14	100	100	100	100	100
2	Valeraldehyde	Almond, Grass, Malt, Oil	12.00	5.74	1.49	2.12	2.54	5.28	3.92
3	Hexanal	Grass, Fat	4.50	100	9.41	12.15	11.77	25.19	16.82
4	2-Methyl-2-butenal		458.90	-	-	-	-	0.14	-
5	Octanal	Fat, Soap	0.70	74.51	-	26.22	19.63	-	-
6	1-1-Nonanal	Rose, Citrus, Fat	1.00	42.68	-	14.11	7.97	-	-
7	1-Pentanol		15.00	-	2.85	-	-	-	5.18
8	1-Hexanol	Potato, Grass, Celery	2.50	34.54	5.20	-	-	-	-
9	1-Heptanol		330.00	0.08	-	-	-	-	-
10	Bicyclo[3.1.1]hept-3-en-2-ol,4,6,6-trimethyl-	Verbena	4.00	-	-	2.67	-	-	-
11	2-Heptanone	Cheese, Fruit, Grass Meat	14.00	2.82	1.81	2.69	1.20	-	4.57
12	2-Nonanone	Fruity, Soap	100.00	-	0.22	0.39	0.20	-	0.49
13	2-Decanone		7.94	3.3	1.00	3.76	3.18	-	4.49
14	Ethyl heptanoate	Pineapple	1.90	27.85	-	-	-	-	-
15	Ethyl caprylate	Pear, Flower and Pineapple	12.87	15.96	0.64	2.06	0.44	-	1.50
16	Octanoic acid	Cheese, Oil, Sweat	5.10	7.73	-	3.57	-	-	-
17	(-)-Limonene	Lemon	10.00	-	-	1.60	3.42	-	-
18	2-Methylpyrazine	Roasted peanut, Nut	60.00	-	0.28	0.27	0.24	-	1.01
19	2,5-Dimethyl pyrazine	Fried peanut, Chocolate	1.80	-	25.69	18.19	10.78	-	52.33
20	Pyrazine,2-methyl-3-(2-methylpropyl)-	Celery	35.00	-	0.28	0.27	0.24	-	1.01
21	Pyrazine,2-ethyl-3,5-dimethyl-	Fried peanut, Coffee	1.00	-	12.83	17.40	8.14	43.18	41.62
22	2-Pentylfurana	Butter, Flower, Fruit	6.00	2.54	-	2.47	1.34	-	-

Aroma characteristics were retrieved from Flavornet. ROAV ≥ 0.1 are presented at least, 0.1 ≤ ROAV < 1: the compound contributed little to the odor, ROAV > 1: the compound is a key volatile compound. “-”: Not identified, or ROAV < 0.1.

## Data Availability

Data is shown in the article.

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
