# Peer review of "Analyzing the Effect of Baking on the Flavor of Defatted Tiger Nut Flour by E-Tongue, E-Nose and HS-SPME-GC-MS"

_foods, 2022, doi:10.3390/foods11030446_

Round 1

Reviewer 1 Report

Analyze the effect of baking on the flavor of defatted tiger nut flour by E- tongue, E-nose, and HS-SPME-GC-MS

  1. Guan; D. Wang; Y. Zhang; Q. Li; and T. Liu

The paper is entitled “Analyze the effect of baking on the flavor of defatted tiger nut flour by E- tongue, E-nose, and HS-SPME-GC-MS.” Guan et al. studied the effect of baking on the volatilome of defatted tiger nut flour.

The manuscript has good data. Nevertheless, it does not answer some critical questions and comments the average reader may have. For example:

  • The authors' setup the baking temperature at 150ºC. Why did they choose only one temperature? Why not run the experiment at two or three different baking temperatures?
  • The metabolomics society pushes GC-MS identifications to be confirmed using reference compounds and not solely reliant on library match. This new criterion may be highly unfair for some researchers since it is challenging to find and acquire 68 standards, as is the case of the authors. Nevertheless, the authors must find a few standards to verify their findings. The authors should prioritize the critical precursors and products of the Maillard and oxidation reactions that they propose in the manuscript.
  • All GC-MS data should be uploaded into a repository. Furthermore, a table similar to Table 1 must be presented with the following information: metabolite name, chemical formula, the retention time, Kovats elusion index, theoretical molecular ion mass (M+), experimental molecular ion mass (if observed), accuracy (PPM), EI-MS fragments (used for identification), matching and reverse matching values, metabolite chromatographic area (indicating if it is normalized or not). In the case of normalization, please detail if it is normalized using the TIC or using only the Sum of identified peaks. Finally, place an asterisk in those metabolites that were verified using standards (see point 2).
  • Finally, is there an Appendix B for the Sensor array of the e-tongue?

Author Response

Response to Reviewer 1 Comments

         First of all, thank you very much for taking precious time to review my paper. I have benefited a lot from your professional guidance on my paper, and I will express my gratitude to you again. Below is the answer to your question and modifications:

Point 1: The authors' setup the baking temperature at 150ºC. Why did they choose only one temperature? Why not run the experiment at two or three different baking temperatures?

 Response 1: In the pre experiment, the reaction rate was moderate when baking at 150 ℃. At this temperature, defatted tiger nut flour with different baking levels can be baked, which can replace the samples that have been subjected to low temperature for a long time and high temperature for a short time, which is representative. Therefore, 150 °C is selected in this paper.

 Based on your valuable comments, the paper has been revised as follows:

 The beginning of the first paragraph of 3.1 in the paper has been changed to“ Reasonable baking can effectively improve the flavor of defatted tiger nut flour, some studies have shown that the Maillard reaction temperature is around 100 °C, and the caramelization reaction temperature is around 120 °C[19]. The baking temperature is too low, it will take a long time to bake, resulting in low baking efficiency and high processing energy consumption; the baking level temperature is too high, it will destroy the nutrients such as amino acids and sugars, the reaction rate is too fast and difficult to be controlled during processing. It is easy to make the sample lose its original odor and produce burnt and bitterness [20]. Therefore, the baking temperature was controlled at 150 °C in this experiment. ”

Point 2: The metabolomics society pushes GC-MS identifications to be confirmed using reference compounds and not solely reliant on library match. This new criterion may be highly unfair for some researchers since it is challenging to find and acquire 68 standards, as is the case of the authors. Nevertheless, the authors must find a few standards to verify their findings. The authors should prioritize the critical precursors and products of the Maillard and oxidation reactions that they propose in the manuscript.

Response 2: Indeed, it is very difficult to find 68 standards. I will study the changes of the types and content of sugar, oil and amino acids in defatted tiger nut flour during baking and the relationship between them and volatile flavor compounds in the next experiment. Because the composition of food raw materials is complex, its types of proteins, amino acids, reducing sugars and lipids are complex, which will have a large and uncertain impact on volatile flavor compounds. In addition, the baking temperature, its own aroma compounds, and the generation method will also affect the generation and transformation of volatile flavor compounds. So it is very difficult to find the critical precursors and products of Maillard and oxidation reactions.

According to your professional opinions, I have consulted a large number of literature on flavor direction, few literature mentioned that a certain precursor component can produce a certain flavor component. Most literature only mention that a large class of components (amino acids, reducing sugars, lipids) can react to produce a large class of components (aldehydes, ketones, alcohols, etc.). According to the valuable opinions of experts and the literature I found, I made the following modifications :

Line 153-155: “3-methylbutanal was produced by the degradation of leucine, which indicats cocoa, sweet and baked aromas increased”

Line 156-158: “The reduction of hexanal indicates that the aroma of soybean, malt and grass is reduced, which is the oxidation product of unsaturated fatty acids such as linoleic acid and oleic acid.”

Line171-172 added “2-heptanone may be broken down by amino acids.”

Line 178-180 added “γ-Butyrolactone is formed after roasting, which has a nutty aroma, resulting from the esterification of hydroxy fatty acids or the oxidation of unsaturated aldehydes.”

Line 181-183 added “As one of the critical products of the Maillard reaction, pyrazines were formed by strecker degradation of leucine, isoleucine and glycine in the reaction”

Line 188-190: “Furans are oxygenated heterocyclic compounds generated by the Maillard reaction and caramelization, mainly from the cyclization and dehydration of Amadori compounds”

Point 3:  All GC-MS data should be uploaded into a repository. Furthermore, a table similar to Table 1 must be presented with the following information: metabolite name, chemical formula, the retention time, Kovats elusion index, theoretical molecular ion mass (M+), experimental molecular ion mass (if observed), accuracy (PPM), EI-MS fragments (used for identification), matching and reverse matching values, metabolite chromatographic area (indicating if it is normalized or not). In the case of normalization, please detail if it is normalized using the TIC or using only the Sum of identified peaks. Finally, place an asterisk in those metabolites that were verified using standards (see point 2).

Response 3: First of all, thank to the expert's very professional guidance, I benefited a lot. I have been paying attention to the research direction of flavor in Foods and other journals, and referring to the list style of volatile flavor compounds in their articles, so I lack some data. Now, according to your professional guidance, I have made the following changes:

(1)I have added the metabolite name, chemical formula, the retention time, Kovats elution index and molecular ion mass (M +) to the Table 1. Chromatographic area added as Appendix D Table 5. The EI-MS fragments have been added as Appendix C.However, there are too many indicators in the table 1, whether some should be reasonably deleted should also be guided by expert.

(2) Due to the matching and reverse matching values are different in different samples. Usually, it can be determined that it is a component only when the matching value >800. The matching value of all VOCs in this paper is more than 800, so I wrote "the result was accepted when matching and reverse matching values were above 800" in manuscript 2.3.

(3) In the case of normalization, my normalization method: quantitative analysis based on area normalization method, remove the erosion of SPME fiber and capillary column, unidentified peak. Only identifiable peaks are used for normalization.

At the end of 2.3, I have revised : "Quantitative analysis based on area normalization method, remove the erosion of SPME fiber and capillary column, unidentified peaks. Only the identified peaks are used for normalization.”

(4) I placed asterisk the metabolites that can be found for standard that were verified using standards in Table 1, such as 3-methylbutanal, 1-hexanal, 2-heptanone, γ-butyrolactone, 2-Pentylfuran.

(5) The accuracy (PPM) may be the detection of certain organic compounds and chemical components in pesticides and other chemicals by LC-MS or GC-MS. I consulted a lot of literature about flavor, but there is no parameter about accuracy (PPM), which may be this parameter is not suitable for flavor identification, so I have not added it for the time being.

Point 3: Finally, is there an Appendix B for the Sensor array of the e-tongue?

Response 4: Thanks to the expert's guidance on details, I have added the sensor array of E- tongue as Appendix B.

    Thanks again for your very professional guidance on GC-MS, I hope my answers and corrections can be satisfactory to you.

Reviewer 2 Report

The manuscript entitled " Analyze the effect of baking on the flavor of defatted tiger nut

3 flour by E- tongue、E-nose and HS-SPME-GC-MS” aimed at describing the effect of baking on the flavor of defatted tiger nut flour by E- tongue, E-nose and HS- 61 SPME-GC-MS

The manuscript is full of naive mistakes. Some examples:

-Line 56-57: “Flavor compounds” should be replaced by “volatile organic compounds (VOCs)” and the acronym “VOCs” should replace some repetitions of “Flavor compounds”, throughout the manuscript;

-Line 131-132: “the baking temperature was controlled at 150 °C” please improve the justify this sentence;

-Line 134: “among which were” should be “among which there were”;

-Line 138-139: “hexanal and 1-Hexanol” at these lines and throughout the manuscript the names of the volatile compounds are reported using sometimes lowercase or uppercase letters. Please, use lowercase letters only;

Lines 142-143: “Aldehydes, mainly produced by lipid oxidation, decomposition, and Strecker

degradation of the amino acid, and its odor threshold is low [23, 24]”: in this sentence the verb is missing.

Lines 144-145: “It accounted for a large proportion of the volatile flavor components of defatted tiger nut flour. It is the main components…”: in this two sentences “It” should be replaced by “They”;

Line 148: “indicating cocoa, sweet and baked…” should be “which indicates that cocoa, sweet and baked…”;

Line 162: “indicating cheese and..” should be “indicating that cheese and..”;

Line 165: “gamma-Butyrolactone” should be “γ-butyrolactone”;

Lines 190-193: The English format is to improve. Please rewrite putting the subjects before verbs;

Line 205: “added pyrazines, added the aroma” should be “produced pyrazines, adding the aroma”;

Line 206: “more” should be “higher”;

Line 231: “As can be seen from..” should be “As it can be seen from..”;

Line 238: “tiger nut flour were mainly pyrazines…” should be “tiger nut flour were mainly composed by pyrazines…”

Lines 280-283: The sentence is not clear. Please, rewrite dividing in two sentences; Lines 180-183: Lines 338-340: The sentence is not clear. Please, rewrite dividing in two sentences;

Line 343: “produced” should be “producing”;

And so on

It is necessary to revise English throughout the manuscript

Author Response

Response to Reviewer 2 Comments

First of all, thank you for taking precious time to review my paper, and providing detailed and professional comments on my paper. According to your guidance, I made the following modifications:

Point 1: Line 56-57: “Flavor compounds” should be replaced by “volatile organic compounds (VOCs)” and the acronym “VOCs” should replace some repetitions of “Flavor compounds”, throughout the manuscript.

Response 1: I have replaced the "volatile flavor compounds" with "volatile organic compounds (VOCs)" in the manuscript abstract Line16. The subsequent volatile flavor compounds is replaced by "VOCs".

Point 2: Line 131-132: “the baking temperature was controlled at 150 °C” please improve the justify this sentence.

Response 2: Line 129-139 have revise this sentence: “ Reasonable baking can effectively improve the flavor of defatted tiger nut flour, some studies have shown that the Maillard reaction temperature is around 100 °C, and the caramelization reaction temperature is around 120 °C[19]. The baking temperature is too low, it will take a long time to bake, resulting in low baking efficiency and high processing energy consumption; the baking temperature is too high, it will destroy the nutrients such as amino acids and sugars, the reaction rate is too fast and difficult to be controlled during processing. It is easy to make the sample lose its original odor and produce burnt and bitterness [20]. Therefore, the baking temperature was controlled at 150 °C in this experiment. ”.

Point 3: Line 134: “among which were” should be “among which there were”.

Response 3: Line140: “among which were” has been changed to “among which there were”.

Point 4: Line 138-139: “hexanal and 1-Hexanol” at these lines and throughout the manuscript the names of the volatile compounds are reported using sometimes lowercase or uppercase letters. Please, use lowercase letters only.

Response 4: The case of compound names in the full text have been corrected. Referring to other literature about flavor, the initials of compound name used uppercase in the table, and  the initials of compound name used lowercase in the text.

Point 5: Lines 142-143: “Aldehydes, mainly produced by lipid oxidation, decomposition, and Strecker degradation of the amino acid, and its odor threshold is low [23, 24]”: in this sentence the verb is missing.

Response 5: Line148-149: I have added verbs to this sentence. This sentence has been modified as follows:“ Aldehydes are mainly produced by lipid oxidation, decomposition, and Strecker degradation of the amino acid, and its odor threshold is low [23, 24]. ”.

Point 6: Lines 144-145: “It accounted for a large proportion of the volatile flavor components of defatted tiger nut flour. It is the main components…”: in this two sentences “It” should be replaced by “They”.

Response 6: Line149-150: In this two sentences, I replaced “It” by “They”, and the two sentences have been modified as follows: “They accounted for a large proportion of the VOCs of defatted tiger nut flour.” and “They are the main compounds that affected the odor of the defatted tiger nut flour. ”.

Point 7: Line 148: “indicating cocoa, sweet and baked…” should be “which indicates that cocoa, sweet and baked…”;

Response 7: Line154: “indicating cocoa, sweet and bake aromas increased” has been changed to “which indicates that cocoa, sweet and baked aromas increased.”.

Point 8: Line 162: “indicating cheese and..” should be “indicating that cheese and..”.

Response 8: Line171: “indicating cheese and..” has been changed to “indicating that cheese and...”.

Point 9: Line 165: “gamma-Butyrolactone” should be “γ-butyrolactone”.

Response 9: Line178: “gamma-Butyrolactone”has been changed to “γ-Butyrolactone”.

Point 10: Lines 190-193: The English format is to improve. Please rewrite putting the subjects before verbs.

Response 10: Line196-197: This sentence has been rewritten as”When baked for 8 min, the relative content of hexanoic acid, octanoic acid, and nonanoic acid increased, indicating coconut and cheese aromas increased...”

Point 11 : Line 205: “added pyrazines, added the aroma” should be “produced pyrazines, adding the aroma”.

Response 11: Line217-218: This sentence has been changed to“The baked defatted tiger nut flour produced pyrazines, adding the aromas of caramel and baked nut. ”

Point 12: Line 206: “more” should be “higher”.

Response 12: Line219: “more” has been changed to “higher”.

Point 13: Line 231: “As can be seen from..” should be “As it can be seen from..”.

Response 13: Line242: “As can be seen from..” has been changed to “As it can be seen from.”.

Point 14: Line 238: “tiger nut flour were mainly pyrazines…” should be “tiger nut flour were mainly composed by pyrazines…”

Response 14: Line248-249: “tiger nut flour were mainly pyrazines…” has been changed to “tiger nut flour were mainly composed by pyrazines…”

Point 15: Lines 280-283: The sentence is not clear. Please, rewrite dividing in two sentences; Lines 180-183: Lines 338-340: The sentence is not clear. Please, rewrite dividing in two sentences.

Response 15: Line301-306: This sentence has been rewrited: “The defatted tiger nut flour without baking and baking for 4 and 8 min were relatively close to each other. Baking for more than 8 min, the distance was further from the unbaked sample, indicating that the baking drastically changed the original odor of the defatted tiger nut flour. The samples baked for 16 and 20 min were relatively close to each other, indicating that their odors were similar, and their sensory odor was mainly burnt and coffee.”.

Line194-197:“Acids were produced by the further oxidation of aldehydes, which have a high odor threshold and have little effect on the odor [40]. They had cheesy, fruity, and sour aromas [41]. When baked for 8 min, hexanoic acid, octanoic acid, and nonanoic acid increased, the aromas of coconut and cheese increased.” has been changed to “Acids were produced by the further oxidation of aldehydes, they have a high o or threshold and have little effect on the odor [40]. They had cheesy, fruity, and sour aromas [41]. When baked for 8 min, the relative content of hexanoic acid, octanoic acid, and nonanoic acid increased, indicating coconut and cheese aromas increased.”

Line359-363:”Tiger nut, an oil-making raw material of excellent quality, produces a large amount of defatted tiger nut flour after oil extraction. Due to its flavor reduction, it is often used as fertilizer and feed [47], the reasonable baking can effectively improve its flavor, realizing a high-value utilization as a food source.” has been changed to “Tiger nut, as a high-quality oil-making raw material, will produce a large amount of defatted tiger nut flour after oil extraction [47]. Due to its flavor reduced , it is usually used as fertilizer and feed after oil extraction [47]. Reasonable baking can effectively improve its flavor, and can be used as high-quality food raw materials to realize its high-value utilization.”.

Point 16: Line 343: “produced” should be “producing”.

Response 16: Line366: “produced” has been changed to “producing”.

    At the same time, due to some naive mistakes in my paper, and based on your and the editor's suggestion, I will also use MDPI's English editing services. Thanks again for your very detailed and professional guidance.

Reviewer 3 Report

Minor remarks

  • All minor remarks are highlighted in the manuscript.

Major remarks

  • Authors should emphasize what is innovative in the manuscript.
  • When writing a manuscript, avoid the first-person plural.
  • Provide a blank space between quantity and unit, only in the case of percentage.
  • References should be prepared according to the instructions for authors.

Author Response

Response to Reviewer 3 Comments

First of all, thank you very much for taking precious time to provide very detailed and professional guidance on my paper, I have made the following corrections:

Point 1: All minor remarks are highlighted in the manuscript.

Response 1: Thank you very much for carefully highlighting the minor comments in the manuscript, I have made the following corrections:

(1)Line44:“and” has been changed to “, and”.

(2)Line69-70: “Tiger nut flour was passed through a 0.42mm (60-mesh) screen, and was added to a tank for supercritical CO2extraction” has been changed to “Tiger nut flour was passed through a 0.42mm (60-mesh) screen and was added to a tank for supercritical CO2 extraction, “.

(3)Line110: “( 25 ± 1 ) °C” has been changed to “( 25±1 ) °C”.

(4)Line117-118: “The defatted tiger nut flour(10g) added 100 mL of 100°C deionized water  and soaked for 30 min. The mixture was centrifuged at 3800×g for 20 min at 20 ◦C, “ has been changed to “The defatted tiger nut flour ( 10g ) added 100 mL of 100 °C deionized water and soaked for 30 min. The mixture was centrifuged at 3800 × g for 20 min at 20 °C, ”.

(5)Line291-292: “The sensors W2S, W1W, W6S, W1S and W5S had significant change,”  has been changed to “The sensors W2S, W1W, W6S, W1S and W5S "had changed significantly,”.

(6)Line374: “can enhanced” has been changed to “ can improve”.

Point 2: Authors should emphasize what is innovative in the manuscript.

Response 2: I have added innovation to Line 61-66:”This experiment is the first time to analyze the taste and odor of defatted tiger nut flour by E-tongue, E-nose and HS-SPME-GC-MS, and screen out the baking conditions reasonably to improve the flavor, and realize its high-value utilization as food raw materials. At the same time, it provides a theoretical basis for the flavor improvement of starchy raw materials such as defatted tiger nut flour after baking.”.

Point 3: When writing a manuscript, avoid the first-person plural.

Response 3: According to the opinions of experts, I have made the following modifications:

(1)Line59: “we” has been deleted.

(2)Line72:“We adjusted the upper and lower heat of the oven to 150°C,”has been changed to ” The upper and lower heat of the oven was adjusted to 150°C, ”.

(3)Line76:We added 0.8g of the sample to the headspace bottle,” has been changed to ”0.8g sample was added to the headspace bottle,”

(4)Line79-80:“. We immediately inserted the gas chromatography inlet,” has been changed to  “, then inserted immediately the gas chromatography inlet,”.

(5)Line109:”We took 1g of defatted tiger nut flour, placed it in a 20mL headspace bottle,” has been changed to “1g defatted tiger nut flour was placed to a 20mL headspace bottle,”.

Point 4: Provide a blank space between quantity and unit, only in the case of percentage.

Response 4: According to the opinions of experts, I have made the following modifications:

(1)Line14: “baking time of >8min” has been changed to “baking time of > 8 min”.

(2)Line72:“150°C”has been changed to “150 °C”。

(3)Line73:“3mm” has been changed to “3 mm”.

(4)Line76:“8g”has been changed to “0.8 g”.

(5)Line77:“50°C”has been changed to “50 °C”.

(6)Line80:“250°C”has been changed to “250 °C”.

(7)Line85-91:  “40°C, 70°C, 3°C/min, 99.99%, ” has been changed to “40 °C, 70 °C,   3 °C/min, 99.99 %” etc.

(8)Line109: “1g,20mL” has been changed to “1 g, 20 mL” 

(9)Line110: “( 25 ± 1 ) °C” has been changed to “( 25±1 ) °C”.

(10)Line117-118: “(10g), 100°C, 3800×g, 20°C, “ has been changed to “( 10 g ) added 100 mL of 100 °C, 3800 × g, 20 °C, ”.

(11)Line120: “25 ± 1 °C” has been changed to“( 25±1 ) °C”.

  (12) Line299-300: “77.7% , 14.9%, 92.6%,”  has been changed to “77.7 %, 14.9 %, 92.6 %,”.

Point 5:References should be prepared according to the instructions for authors.

Response 5: According to the expert opinion, I have carefully checked the references and made the following modifications:

(1)Line29-30: The position of reference 1 in the text has been changed“Tiger nut (Cyperus esculentus L.), also known as wild chestnut, underground walnut, etc. [1],originated inNorth Africa and the Mediterranean, and is widely cultivated in Spain, Italy, South Africa, and other places [2]. It is a highly-efficient and high-quality edible oil crop that is drought-enduring and barren-resistant.”has been changed to“Tiger nut (Cyperus esculentus L.), also known as wild chestnut, underground walnut, etc. It is a highly-efficient and high-quality edible oil crop that is drought-enduring and barren-resistant [1]. It originated in North Africa and the Mediterranean, and widely cultivated in Spain, Italy, South Africa, and other places [2]. ”

(2) The format of reference 14 has been modified.”Laureati M., Buratti S., Bassoli A., et al. Discrimination and characterisation of three cultivars of Perilla frutescens by means of sensory descriptors and electronic nose and tongue analysis[J]. Food Research International, 2010, 43(4).” has been changed to “LAUREATI,;BURATTI, S.; BASSOLI, A.; BORGONOVO, G.; PAGLIARINI, E. Discrimination and characterisation of three cultivars of Perilla frutescens by means of sensory descriptors and electronic nose and tongue analysis[J]. Food Research International, 2010, 43, 959-964.”

(3)The original reference 15 was deleted.

(4) Reference 18 has been changed. “Cai,C.; Tang, F.X.; Guo, Z. Effects of pretreatment   methods and leaching methods on jujube wine quality detected by electronic senses and HS-SPME–GC–MS[J]. Food Chem, 2020, 330.” has been changed to “Cai, W.C.; Tang, F.X.; Guo, Z. Effects of pretreatment methods and leaching methods on jujube wine quality detected by electronic senses and HS-SPME–GC–MS[J]. Food Chem, 2020, 330, 127330.”

(5) References 19 “SONG, Y.; DU, B.; DING, Z.; YU, Y.; WANG, Y. Baked red pepper

(Capsicum annuum) powder flavor analysis and evaluation under different exogenous Maillard reaction treatment. Lwt. 2021, 139, 110525.” has been changed to “YE, T.T.; LIU, J.; WAN, P.; LIU, S.Y.; WANG, Q.Z.; CHEN, D.W. Investigation of the effect of polar components in cream on the flavor of heated cream based on NMR and GC-MS methods[J]. LWT, 2022, 155.”

(6) Reference 21 was addedLine146:“Li, X.Z.; LIU, S.Q. Effect of pH, xylose content and   heating temperature on colour and flavour compound formation of enzymatically hydrolysed pork trimmings[J]. LWT, 2021, 150.”

(7) Reference 26 was addedLine158:” YIN, X.Y.; LV, Y.C.; WEN, R.X.; WANG, Y.; CHEN, Q.;   KONG, B.H. Characterization of selected Harbin red sausages on the basis of their flavour profiles using HS-SPME-GC/MS combined with electronic nose and electronic tongue[J]. Meat Science, 2021, 172, 108345.”

(8)The original Reference 31 is deleted.

(9) Reference 34 was added Line183:” SPADA, F.P.; BALAGIANNIS, D.P.; PURGATTO, E.; ALENCAR, S.M.; CANNIATT-BRAZACA, S.G.; Parker, J.K. Characterisation of the chocolate aroma in roast jackfruit seeds. Food Chem. 2021, 354, 129537.”

Thanks again for your very professional and careful guidance, I wish you good health and smooth work!

Round 2

Reviewer 1 Report

Please do not forget to include a link to the repository where the GC-MS data is uploaded.

This manuscript is a resubmission of an earlier submission. The following is a list of the peer review reports and author responses from that submission.